# A Pilot Randomised Controlled Trial of a Text Messaging Intervention with Customisation Using Linked Data from Wireless Wearable Activity Monitors to Improve Risk Factors Following Gestational Diabetes

**DOI:** 10.3390/nu11030590

**Published:** 2019-03-11

**Authors:** N. Wah Cheung, Caron Blumenthal, Ben J. Smith, Roslyn Hogan, Aravinda Thiagalingam, Julie Redfern, Tony Barry, Nancy Cinnadaio, Clara K. Chow

**Affiliations:** 1Department of Diabetes & Endocrinology, Westmead Hospital, Westmead NSW 2145, Australia; Roslyn.Hogan@health.nsw.gov.au (R.H.); cinnadaio@gmail.com (N.C.); 2Westmead Clinical School & Westmead Applied Research Centre, Faculty of Medicine and Health, University of Sydney, Westmead NSW 2145, Australia; Aravinda.Thiagalingam@health.nsw.gov.au (A.T.); julie.redfern@sydney.edu.au (J.R.); clara.chow@sydney.edu.au (C.K.C.); 3Division of Women and Newborn Health, Westmead Hospital, Westmead NSW 2145, Australia; Caron.Blumenthal@health.nsw.gov.au; 4School of Public Health, University of Sydney, Sydney NSW 2006, Australia; ben.smith@sydney.edu.au; 5Department of Cardiology, Westmead Hospital, Westmead NSW 2145, Australia; tonybarry@mac.com

**Keywords:** gestational diabetes, lifestyle program, M-health, text messaging, activity monitor, diabetes prevention, randomized controlled trial

## Abstract

Gestational diabetes (GDM) is a highly prevalent disorder of pregnancy which portends a high risk for future type 2 diabetes. Limited evidence indicates lifestyle intervention prevents the development of diabetes, but most previously studied interventions are resource-intensive. Intervention programs that utilise newer technologies may be scalable at lower cost. This 6-month pilot randomized controlled trial tested the delivery of text messages linked to an activity monitor, adaptive physical activity goal setting, and limited face-to-face counseling, as an intervention to improve rates of post-partum glucose tolerance testing and lifestyle behaviours amongst women following a GDM pregnancy. Sixty subjects were randomised 2:1 intervention vs. control. Compared to control subjects, there were trends for intervention subjects to improve diet, increase physical activity, and lose weight. There was no difference between the groups in the rate of glucose tolerance testing. Only 46 (77%) subjects completed some, and 19 subjects completed all the elements of the final evaluation. Feedback regarding the text messages and activity monitor was highly positive. Overall, results suggest that a text message and activity monitor intervention is feasible for a larger study or even as a potentially scalable population health intervention. However, low completion rates necessitate carefully considered modification of the protocol.

## 1. Introduction

The International Diabetes Federation has estimated that globally, gestational diabetes (GDM) affects some 14% of pregnancies [1]. Although it resolves at the end of the pregnancy, about half of these mothers will develop diabetes in the future, a rate that is 3.5 to 7 times that of women who did not have GDM in pregnancy [2,3,4]. The population attributable risk of GDM for the development of diabetes has been estimated to be between 10% to 31%; that is up to one in three women who have type 2 diabetes may have had an earlier GDM pregnancy [2]. Yet despite this heightened risk for diabetes, there is a gap in preventative management. Surveys indicate women with prior GDM are more sedentary and have poorer diet compared to the general population [5,6], and also that attendance for the recommended post-partum glucose tolerance test (GTT), to confirm resolution of GDM, is poor [7,8]. This may lead to delayed diagnosis of diabetes, and also importantly, failure to undertake planning for type 2 diabetes in pregnancy prior to the next pregnancy [9].

Intensive lifestyle intervention programs directed at people with impaired glucose tolerance (IGT) can reduce the progression to Type 2 diabetes (T2DM) [10,11]. These interventions have been shown to be effective in the subgroup of women with IGT who have had a history of prior GDM [12]. There is also much interest in developing programs for women in the period after the GDM pregnancy, well before the onset of IGT. This is an opportune time to implement diabetes prevention strategies, as these women at risk are identified and in contact with the health system. However, studies performed to date, using structured behavior change strategies, and intensive face-to-face dietary and lifestyle sessions, have had mixed success in demonstrating improvements in physical activity, dietary behavior or weight management [13,14,15,16,17,18,19,20,21,22]. Low participation rates and high dropout have been major challenges for studies performed in this population. The need for childcare, return to work, and low self-efficacy have been major barriers [23,24]. Furthermore, any resource intensive program which requires multiple or regular face-to-face visits is not practical or sustainable for a problem which is now a major public health issue, particularly for women who are time poor with young families or have limited access to lifestyle programs.

Novel interventions, using new technologies for delivery of programs requiring only modest resources, need to be explored. M-health technology refers to medical and public health practice supported by mobile devices such as mobile phones, wearable monitoring devices, personal digital assistants and other wireless devices [25]. The proliferation of m-health technologies present options for delivering health interventions on a large scale, using equipment which is already owned and utilised by many people. Women who have GDM are young and generally high-end users of technology. Most have smart phones and regularly use the internet [26]. They are busy and do not have time to attend conventional health services for preventative health care. They should be an ideal population to target using mobile technology to improve health care and behaviour.

We examine the feasibility of an intervention comprising an individualised mobile phone text-messaging education and support program, additionally customised through linking with a wearable activity monitor, in a randomized controlled trial (RCT) to improve participation in post-partum glucose tolerance testing and health behaviours amongst women with recent GDM.

## 2. Materials and Methods

### 2.1. Study Design

SMART MUMS WITH SMART PHONES was a single centre open-label pilot RCT. The intervention was designed to build on a text messaging intervention (TEXT ME), with messages customised for individual subjects, which demonstrated effectiveness in reducing cardiovascular risk among people with coronary artery disease [27]. The TEXT ME program was adapted to provide appropriate messaging to women in the period after a GDM pregnancy, to improve lifestyle behaviour. In addition, the program was supplemented by the use of a wireless and wearable activity monitor that both enabled tracking of activity and further customisation of the text messages that were sent. The primary aim was to establish the feasibility of the intervention and protocol implementation with the goal that pilot data generated would contribute to the development of a large scale RCT. Ethics approval was obtained from the Western Sydney Local Health District Research Ethics Committee. The trial was registered with the Australian New Zealand Clinical Trials Registry ACTRN12616000067471 and conducted in accordance with the Declaration of Helsinki. All participants gave written informed consent.

### 2.2. Participants and Randomization

The study planned to recruit 60 women to be randomised in a 2:1 ratio to receive the digital health support program (40 participants) or to receive standard paper-based information only (20 participants). Randomisation was undertaken by computer random number generation, using a permuted block size of 4. Inclusion criteria were: GDM, as diagnosed by the 1998 Australasian Diabetes in Pregnancy Society Criteria [28] which required a fasting glucose level ≥5.5 mmol/L and/or a 2-h glucose level ≥8 mmol/L on a 75 g oral glucose tolerance test, age ≥18 years; owning a smart mobile phone with text messaging capability; having internet access; having sufficient skill in the English language to read text messages, and being physically capable of performing moderate intensity physical activity. Our tertiary level hospital cares for about 900 women with GDM per annum. All women in our health district who choose to be treated in a public hospital for free attend our clinics. In 2018, 83% of Australian adults owned a smart phone [29], and the vast majority of women attending our clinics own smart phones. Women meeting eligibility criteria were approached by the study coordinator when attending our dedicated Diabetes in Pregnancy antenatal clinic. Women attending this clinic were largely those who required insulin, as women not needing insulin were seen in regular antenatal clinics. Women were recruited whilst 24–30 weeks pregnant, in order to collect baseline data and to enable them to experience use of the activity monitor before they were more constrained by advanced pregnancy. The study protocol and consort diagram are outlined in Figure 1.

### 2.3. Intervention

The intervention was a digital health support program involving education and support delivered via customized mobile phone text messages, with the use of an activity monitor that was integrated with the texting through the use of activity data. The use of the activity monitor further customised text messages through a detailed and responsive algorithm. The program was complemented by two 30 min diet counselling sessions—a face-to-face session in clinic at initial recruitment to the program when pregnant (over and above standard GDM care) and a repeat session by phone at 10–12 weeks post-partum. These sessions were delivered by a dietitian and focused on the adoption phase of behavior change, as tested in our previous post-GDM lifestyle intervention programs [14,15], promoting Australian dietary guidelines [30] after pregnancy, and building physical activity into one’s daily routine, with the aim of achieving 10,000 steps a day. The repeat post-partum counselling session reinforced the importance of returning to pre-pregnancy weight, controlling carbohydrate intake, and the use of low carbohydrate vegetables and low carbohydrate foods to satiate hunger. They also served to remind subjects in the intervention group of the text messages, and of the use of the activity monitor.

### 2.4. Activity Monitor

Participants were given a Fitbit Flex^®^ activity monitor to assist self-monitoring of physical activity. These monitors were linked to our digital platform, enabling the data to be integrated with our texting program. The Fitbit Flex^®^ has a wrist band for easy wear and an accompanying mobile phone app which enable data to be synchronised from the activity monitor to the phone. The Fitbit Flex^®^ has a limited display showing a progress bar to achieving the daily step target, but the phone app displays information such as number of steps taken, the daily step target, and hours of sleep. The Fitbit Flex^®^ needed to be recharged every 3 days and synchronised every week with a smart phone for the data to be uploaded. Women were shown how to use the Fitbit Flex^®^, charge, and synchronise it at the baseline clinic visit. They were given assistance to download the app to their phone. 

### 2.5. Text Messages

The text messaging program commenced 3 weeks post-partum, and extended for 26 weeks after the post-partum counseling session at 10–12 weeks. Existing validated messages used in the TEXT ME trial [27] were reviewed for use in the SMART MUMS WITH SMART PHONES study. Additional messages, appropriate for this population of young mothers and in accordance with local or national guidelines, were developed by a team which included experts in diabetes, nutrition, physical activity, health promotion, and lactation. The process followed was similar to that applied to the TEXT ME program [31], whereby the working group initially developed the text messages, then these were reviewed for readability and to ensure messages were presented with a positive focus. The text messages then underwent testing with feedback and further modification. Examples of final text messages include: “Small steps count! Just 10 min sessions count towards your target of 30 mins per day. Keep active!”; “Most of your meal should be salad or cooked green vegetables. Fill more than half of your plate with these foods”; “Did you know that breast-fed children get fewer allergies?”; “Reduce or avoid ghee in your cooking for a healthier meal!”.

Text messaging was managed by the TEXT ME engine we have used in earlier studies [27]. This delivers messages in accordance with prespecified algorithms using patient baseline data entered into the message management system. Earlier messages focusing on newborn health and early motherhood, and later messages focusing on achievement and maintenance of physical activity and healthy eating, as well as diabetes prevention. From week 3 post-partum, 3 messages a week, were delivered, one from each of 3 message banks, these being (1) physical activity, (2) nutrition, (3) general health and motherhood information and education. The text messages provided advice, motivation, information and support with a focus on addressing and overcoming known barriers to behaviour change. The messages were semi-personalised, with customisation to cater for women who were not breastfeeding, or were South Asian (for the large South Asian population attending our clinic). There were 2 reminders to undertake glucose tolerance testing within the first 10 weeks post-partum.

From weeks 6 post-partum an additional activity monitor related messages was sent. An automatic computerised algorithm utilised data uploaded from the activity monitors to generate this message, which gave adaptive step targets and feedback regarding their activity monitor activity.

Messages were sent until week 36–38 post-partum (26 weeks from the baseline evaluation and counseling session which occurred at 10–12 weeks). A log was kept of contacts that the subjects made with the study team, and the reason for contact. If required, intervention subjects were given training on how to read, delete and save text messages.

### 2.6. Activity Monitor Related Messages with Adaptive Daily Step Targets

We used the activity monitor to generate customised messages to provide women with weekly adaptive step targets, encouragement, and reminders based on their activity monitor data. For the first 10 weeks post-partum, the daily step target was set at 3500. Adaptive step targets were set weekly based on a rank order percentile algorithm, using the step counts from the previous 2 weeks (Figure 2). An incremental daily step target was set each week by adding 500 steps to the 60th percentile of the steps in the previous 2 weeks. The use of the 60th percentile for adaptive goal setting has been successful in research using pedometers to facilitate weight loss, and such a schedule of reinforcement has been demonstrated to be effective in shaping behavior change [32,33]. When subjects had a decrease in the 60th percentile daily step count, the target from the previous week was retained. The maximum target was 10,000 steps a day. Where the activity monitor had not been synchronized for more than 5 days, a text message would remind women to do so.

In addition to the text message informing women of their weekly step target, the adaptive step target was visible to them on their phone app. Women were encouraged to view their phone app frequently.

### 2.7. Control Subjects

Control subjects received the booklet “Life after Gestational Diabetes” only [34]. This was developed by Diabetes Australia to provide healthy lifestyle advice and information to women who have had GDM. They were contacted for study procedures and evaluations but otherwise had had no interaction with the investigators. 

### 2.8. Study Outcomes and Measurements

The outcomes of the study were
(i)attendance for the post-partum GTT within 12 weeks post-partum;(ii)whether physical activity recommendations (30 min of moderate intensity physical activity at least 5 days a week, measured by self-report and 10,000 steps per day by pedometer count) were met at 6 months;(iii)whether dietary macronutrient recommendations for fat and fibre intake (dietary fat ≤30% of total calories, saturated fat <10%, 15 g of fibre per 1000 calories) were met at 6 months and;(iv)the change in self-reported weight at 6 months

Additional data collected included information on breast feeding, and subject evaluation of the intervention program including acceptability of the SMS messages and the messaging system, and feedback on the use of the activity monitor and its app.

### 2.9. Trial Procedures

Following recruitment, all women underwent an initial face-to-face visit during the pregnancy where they completed the Active Australia Questionnaire (AAQ) [35] regarding their pre-pregnancy activity, and a 24-h food recall. At 10–12 weeks post-partum the baseline evaluation visit was conducted. The final evaluation at the end of the program was conducted at 36–38 weeks post-partum. The 2 evaluations were conducted by telephone. At each evaluation visit, the following data were collected: breastfeeding practice, self-reported weight, AAQ, 24-h food recall. At the 10–12 week post-partum evaluation, women were also asked if they had undertaken their post-partum glucose tolerance test, and the results of these were obtained. For the women who had developed diabetes, clinical management was left with the woman’s primary practitioner, and the research team did not become involved. For the final evaluation visit, all participants were also asked to wear a validated pedometer (Yamax Digi-walker SW700^®^, Yamasa Tokei Keiki Co., Tokyo, Japan), for a period of one week, and complete a 3-day food record in addition to the 24-h food recall.

### 2.10. Dietary and Physical Activity Measurements

The 24-h food recalls and AAQs were undertaken by an experienced dietitian. All women were given measuring instruments to facilitate the accurate recording of the 3-day food record, and these were returned by mail. Baseline evaluation data was based on the 24-h food recall only, but the final evaluation included data obtained from both the 24-h food recall and the 3-day food record. Nutrient analysis was performed using FoodWorks 8 with the AUSNUT2013 database (Xyris Software, Spring Hill, Australia). The estimated dietary energy requirement at the baseline evaluation was estimated utilizing Australian and New Zealand Nutrient Reference Value tables [36], adapted for pregnancy.

The AAQ records self-reported walking, moderate and vigorous intensity physical activity for a one week period. Total activity time was calculated by adding these 3 parameters, but with a doubling of the vigorous activity time [35].

Yamax Digi-walker SW700^®^ pedometers were sent to all participants with a logbook close to their final evaluation at 36 weeks post-partum. The participants were instructed to record the step count at the end of the day, for 7 consecutive days, as well as the duration of time that it was worn. The pedometers were reset at the end of each day. The pedometers and logs were returned by mail.

### 2.11. Process Measures and Message Feedback

At the time of the final evaluation, women in the intervention arm were asked to complete a printed questionnaire. This gave the opportunity for the women to provide feedback regarding the text messages and the use of the activity monitor. This was returned by mail.

Data from the activity monitors was recorded whenever women synchronised them to the Internet. This enabled analysis of the regularity of use of the activity monitors as well as the daily step count.

### 2.12. Statistical Analysis

Analysis was by intention to treat using SPSS version 22 (IBM Corporation, NY, USA, 2013). Data were analysed by chi-square testing (Fisher’s exact test) for categorical variables or t tests for continuous variables. *p* values of <0.05 were considered significant. Sample size calculations were undertaken using StatTools [37].

## 3. Results

Sixty subjects were recruited into the study, with 40 randomised to intervention and 20 to control. Baseline characteristics of the subjects were well matched (Table 1). Only 46 (77%) women undertook some elements of the 6-month evaluation, and 27 women (19 intervention, 8 control) completed all the elements.

### 3.1. Primary Outcomes

Study outcomes are listed in Table 2. There was no difference in the likelihood of having a post-partum GTT between the control and intervention groups at 12 weeks (65% vs. 70%) (Table 2). One woman from each arm of the study had developed diabetes on the GTT. The glucose values on the GTT were not different between intervention and control subjects (fasting glucose 4.9 ± 0.7 vs. 5.0 ± 4.2 mmol/L, *p* = 0.9; and 2 h glucose 7.2 ± 2.2 vs. 6.4 ± 2.0 mmol/L, *p* = 0.2). There was also no difference between the control and intervention groups in terms of dietary and physical activity targets achieved. However, intervention subjects had a higher fat and lower carbohydrate intake, as percentages of total energy intake (Table 2).

### 3.2. Feasibility for a Future Study

#### Text Message Feedback

Twenty six subjects in the intervention group gave feedback regarding the text messages (Table 3). The vast majority found the messages were helpful, though the reported effects on diet and physical activity were more modest.

There were 228 return text messages sent from intervention subjects to the message engine. Ninety eight of these were responses to the regular messages sent to them (e.g., “thank you”, “OK”), 85 were related to the activity monitor (e.g., activity monitor not working, activity monitor strap broken), 27 related to the study or the evaluation visits, and 18 were related to other issues.

### 3.3. Activity Monitor Feasibility and Utilisation

We evaluated utilisation of the activity monitor in the intervention group by determining the number of days that data was synchronised and uploaded, and therefore available, for the 6 months from 10 weeks post-partum. There were 6 intervention subjects for whom there was no data, suggesting that the activity monitor was either not worn or not synchronized at all. Amongst the subjects for whom there was data, data was available for 47 ± 33% of the days. Fifteen subjects had activity monitor data for at least half the days and 21 subjects had data for at least 28 days. For subjects who wore the activity monitor for at least 28 days, the mean daily step count was 4660 ± 4081 (median 3929, interquartile range 2030–5450). There were 2 subjects who averaged over 10,000 steps a day over the course of the study.

Analysis was conducted to ascertain if there were baseline factors which predicted activity monitor use (Table 4). We considered subjects for whom activity monitor data was available on at least 50% of the days as being regular users (*N* = 15), and <50% of the days as irregular users (*N* = 25). Age was dichotomised at the 50th percentile to generate 2 equal sized groups for this analysis. There were no factors which predicted activity monitor use. Regular activity monitor users were not more likely to meet physical activity guidelines pre-pregnancy.

During the course of the intervention, 27 intervention subjects had at least one problem related to the activity monitor. There were 4 occasions when subjects lost their activity monitors, 18 occasions where the wristband or clip was lost or damaged, 6 when the charger was lost, and 26 where the activity monitor required a factory settings or password reset. Although all lost or damaged items were replaced, these mishaps resulted in disruption to the study and periods where the activity monitor was not worn.

The majority of respondents in the intervention arm provided positive feedback regarding the activity monitors (Table 5). Five subjects expressed concerns regarding the reliability and durability of the Fitbit^®^ and its system.

### 3.4. Study Engagement

To examine study engagement, we examined whether any specific factors predicted completion of the study, defined as undertaking all elements of the final evaluation. Subjects who failed to complete all elements of the evaluation were more likely to be of South Asian ethnicity (*p* = 0.04) or primiparous (*p* = 0.005) (Table 6).

### 3.5. Sample Size Calculations for a Future Study

The results of the physical activity measures, dietary energy intake, and body weight were utilised to calculate sample sizes required to demonstrate an improvement with the intervention. On the basis of the AAQ, and using standard deviations for the entire study population, a total sample size of 378 subjects is required for 80% power at a significance level of 0.05 to demonstrate a difference in total weekly activity time. The sample size requirements for other key outcome variables are listed in Table 7.

Sample size requirements for 80% power to achieve significant differences in principal outcomes in a larger study, based on results of the SMART MUMS WITH SMART PHONES study. Standard deviations from the entire cohort at the final visit were used to calculate sample size.

## 4. Discussion

This pilot randomized controlled trial of a digital health support intervention comprising text messaging linked with activity monitors demonstrated that the intervention program was able to be implemented and that many participants found the program to be appealing. The main challenges of protocol implementation were low compliance and a high drop-out rate, in terms of utilisation of the activity monitor, completion of follow-up data at the 6-month evaluation, but there was high acceptance of the text messaging program. There were non-significant trends towards an improvement in levels of physical activity and dietary intake, with no improvement in achieving targets, but the study was minimally powered to examine these outcomes. 

In the Diabetes Prevention Program, it has been demonstrated that intensive lifestyle interventions can reduce the incidence of diabetes by half amongst women with IGT who have had GDM [12]. RCTs specifically targeting women after a GDM pregnancy have been less successful. The largest study, a 6-month cluster RCT with 2280 women, found that a telephone coaching intervention facilitated weight loss and increased physical activity but there were no differences in dietary intake or rates of diabetes and prediabetes [17]. Several small studies have also targeted behavioural lifestyle changes with interventions based around telephone, group or individual sessions [13,14,15,16,19,20,21,22]. Some have shown improvements in dietary measures [15,16], others have found increases in physical activity, and some have reduced weight [18,20,21]. However, these studies have not been able to demonstrate improvements in glycaemic parameters, or lacked the power to do so. The interventions applied in these studies were resource intensive and at best, were of modest effectiveness.

With GDM now being a highly prevalent disorder which identifies women at risk of future diabetes, it is essential to identify scalable, sustainable interventions to promote preventive health behaviours. One forward looking study tested a structured web-based program with feedback from pedometer data, but unfortunately failed to show improvements in anthropometry, physical activity or glycaemic measures [38]. In our program, we adopted 2 forms of m-health technology, text messaging and wireless web-linked wearable activity monitors, and linked them to test a novel intervention which supports lifestyle change amongst women with recent GDM.

Text messaging has been demonstrated to be an effective means of achieving behavior modification in some situations. In particular, it has successfully supported smoking cessation, and in some studies has increased physical activity and weight loss. Text messaging based interventions have also been shown to improve clinical parameters such as blood pressure, cholesterol, and glycated haemoglobin in people with coronary artery disease or diabetes [27,39]. With an automated delivery system, text messages can be rapidly and inexpensively delivered to large numbers of subjects. With the widespread adoption of smartphones, this is an ideal means of providing reinforcement messages for behaviour change in a population, without requiring the considerable resources and time needed for face-to-face interventions. Having systems for customising the messages enables messages to be personalised, and made culturally and clinically appropriate for the recipient.

Pedometers have been used to measure and support health interventions for many years, and accelerometers have been used in research settings. However, the advent of commercially available wireless wearable activity monitors which link to electronic devices such as mobile phones has led to an explosion in their use. These are convenient, and the utilisation of smart phone applications increases opportunities for feedback of detailed metrics to the wearer, enhancing the experience. Hence, there has been growing interest in the use of modern wearable activity monitors to promote fitness and wellness, but also potentially as health interventions. A systematic review of wearable activity monitor interventions, including pedometers, found that passive interventions, where the user did not also receive any other support, were not effective in promoting behavior change [40]. However, when wearable activity monitors were paired with behavioural change techniques (such as counseling, goal setting, motivational phone calls and text messaging), a number of studies demonstrated improvements in physical activity or weight management. Research into the health benefits of modern wireless wearable activity monitors remains sparse and is lagging behind their widespread adoption as a tool for supporting a healthier lifestyle [41].

Our study demonstrated a trend to improvement in physical activity, dietary energy intake and body weight with the linked combination of text messaging and wireless wearable activity monitors. There was a shift from the higher carbohydrate requirements of pregnancy, to a more modest carbohydrate intake post-partum with a corresponding increase in fat intake, and a non-significant reduction in total energy intake. Feedback suggests that simplification or a reduced need for charging and synchronization of the activity monitor, as well as greater durability of the wrist strap might improve usage and step data collection. A more comprehensive visual display may improve feedback to the women. Newer commercially available wireless wearable activity monitors do overcome some of these issues. Engagement might be improved through more frequent telephone sessions with a lifestyle coach [42], and video conferencing applications are now used by some dietitians to provide home-based consultations. Whilst these m-health technologies should be considered in developing programs to reduce diabetes risk amongst young women who have had GDM, introducing these additional elements into an intervention program will have resource implications and limit its scalability.

The literature suggests that attendance for post-partum GTTs after GDM is poor despite recommendations to undertake this. In an American survey, only 33% of women reported post-partum glucose testing [7]. In a post-partum survey in Australia, 73% of respondents indicated that they had some form of glucose testing (including fingerstick testing), of whom 27% had this done within 8 weeks post-partum [8]. There was only a 36% response rate to the survey so the true incidence is likely to be even lower. Postal reminders have been shown in one study to increase post-partum rates of glucose tolerance testing [43]. However, as in the case of our trial, an earlier Australian RCT of SMS reminders failed to increase testing rates [44]. Interestingly high testing rates of 65–77% were observed among the control subjects in both studies. In Australia, since 2011, women with GDM registered with the National Diabetes Services Scheme have received postal reminders to undertake a post-partum GTT. This may have already encouraged women responsive to reminders in both RCTs to undertake testing, with text message reminders having no additional effect. 

The low completion rate in our study is a concern for future studies in this area. Other studies have reported similar challenges with low engagement or exposure to the full intervention and poor completion rates in the post-GDM population [14,15,21,22]. The demands of work and childcare, lack of social support and access to care and advice, financial difficulties and cultural expectations have all been cited as significant barriers to participation in healthy living programs and behavior [5,22,23,24,45]. The current study additionally noted that being primiparous was predicted of low engagement in the study. It is possible that women who have already had children have learnt to cope better with the demands of having a newborn and were therefore able to better maintain engagement despite having more than one child. We found that women of South Asian background were less likely to complete the study, and greater dropout from a study of wearable activity monitors has also been observed amongst people of Indian ethnicity in Singapore [46]. Whilst we tailored the dietary text messages for South Asian women, greater consideration needs to be given to cultural issues in general, to improve engagement. The high attrition rate in the use of activity monitors is also a challenge, with only 10% of participants in one RCT still wearing their device at 12 months [46].

Our pilot study has demonstrated that a healthy lifestyle intervention based around text messaging and activity monitors is suitable for some women with recent GDM, but major protocol modifications are required to improve completion rates for a larger trial to be feasible. A number of lessons have been learnt which may facilitate the development of a future digital health support program. More consumer engagement with women from a range of backgrounds who have had recent GDM may assist with the development of message content, frequency, and against other commitments of being a new mother. An even greater focus on support and emotional well-being, may be helpful as women may become frustrated by health care providers telling them what to do, without being aware of their full circumstances [45]. Further customisation, perhaps based on feedback during the course of their participation, may help refine the text-messages delivered to each individual to suit their circumstances. Further qualitative research is needed to explore what might be more effective. As m-health technologies evolve, new options such as activity monitors with more displays and other capabilities, and the greater use of the internet, telephone or video-conferencing to support the women should be explored. Simplification of data collection is important.

It is also critical to recognise that the problem of low completion rates in a study environment does not necessarily indicate that a diabetes risk reduction program based around text messaging and activity monitors would be unsuccessful in practice. Our definition of non-completion included the failure to collect all data elements for the 6-month evaluation. The feedback we received regarding the text messages, even from some subjects whom we classified as non-completors, was positive, and the majority did utilize their activity monitor, albeit intermittently. If we consider the intervention as a public health program rather than a clinical intervention, small changes undertaken by many individuals may significantly influence population health.

The repeatedly documented challenges of conducting behavioural intervention trials for this population and collection of high quality data, along with the large sample size and long duration of follow-up required to demonstrate a reduction in the rate of development of diabetes, raises the question as to whether further trials should be undertaken, or whether we should move directly to translate such a program into policy and practice. The cost of text messaging is minimal and even the cost of activity monitors is modest. M-health technologies can be harnessed to develop scalable and economically viable intervention programs. The implication of GDM for the future development of type 2 diabetes is now a major population health issue, and its management should be considered in this context.

## Figures and Tables

**Figure 1 nutrients-11-00590-f001:**
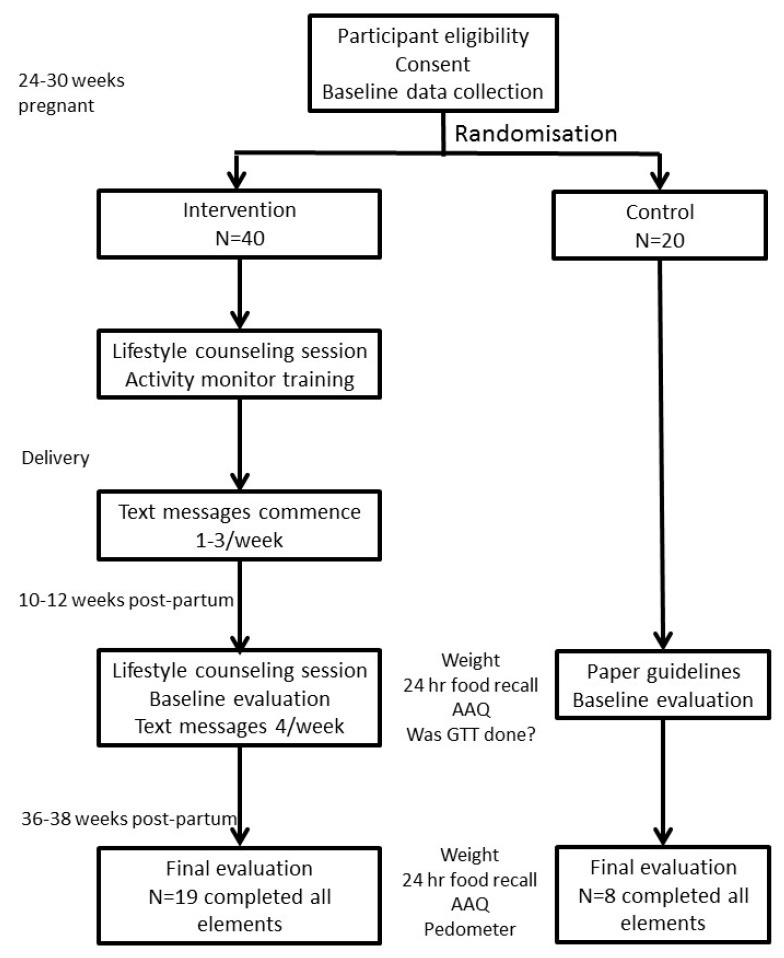
Study protocol and consort diagram.

**Figure 2 nutrients-11-00590-f002:**
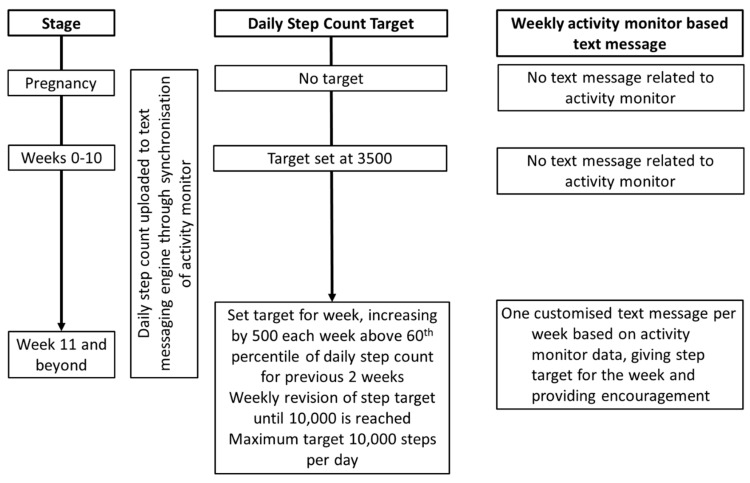
Algorithm for setting adaptive step targets. The step target was set on a weekly basis, based on data uploaded from the activity monitor, and the woman informed through text messaging.

**Table 1 nutrients-11-00590-t001:** Subject baseline characteristics. Data are presented as mean ± SD or proportion (%).

Variable	Intervention (*N* = 40)	Control (*N* = 20)	*p* Value
Age	34 ± 4	34 ± 4	0.90
First degree family history of diabetes	21/40 (53%)	11/20 (55%)	1.0
Pre-pregnancy BMI (kg/m^2^)	28.7 ± 7.6	27.8 ± 4.2	0.60
Pregnancy GTT (glucose level at *T* = 0, mmol/L)	5.2 ± 0.8	5.0 ± 6.5	0.45
Pregnancy GTT (glucose level at *T* = 120, mmol/L)	9.0 ± 1.4	9.0 ± 1.3	0.95
Insulin use during pregnancy	29/40 (73%)	17/20 (85%)	0.35
Gestation (weeks)	38.9 ± 1.2	38.7 ± 0.8	0.44
Birthweight (g)	3235 ± 475	3487 ± 561	0.07
Region of birth			0.58
South Asia	22	12	
Southeast Asia	8	2	
Australia	6	5	
Others	4	1	
Daily energy intake during pregnancy (kJ)	9212 ± 2070	8858 ± 2461	0.56
Daily fat intake (% total energy)	33 ± 9	30 ± 7	0.19
Daily saturated fat intake (% total energy)	10 ± 6	9 ± 3	0.35
Daily carbohydrate intake (% total energy)	43 ± 10	47 ± 8	0.34
150 min Mod PA pregnancy	20/40 (50%)	13/20 (65%)	0.41
Prepregnancy total activity time (min)	176 ± 145	235 ± 228	0.22

**Table 2 nutrients-11-00590-t002:** Study Outcomes. Data are presented as mean ± SD or proportion (%).

Outcomes	Intervention (*N* = 40)	Control (*N* = 20)	*p* Value
**Primary Outcomes**			
GTT performed by 12 weeks post-partum	28/40 (70%)	13/20 (65%)	0.77
Diet fat intake <30% of total energy at final evaluation	4/24 (17%)	4/11 (36%)	0.23
Sat Fat <10% of total energy at final evaluation	10/24 (42%)	6/11 (55%)	0.72
Dietary fibre >15 g/1000 cal at final evaluation	6/24 (25%)	6/11 (55%)	0.13
Achieved 150 min/week moderate intensity activity at final evaluation	11/29 (38%)	5/13 (38%)	1.0
Achieved mean of 10,000 steps a day at final evaluation	1/26 (4%)	0/11 (0%)	1.0
Weight change between final evaluation and 10–12 weeks post-partum (kg)	−1.7 ± 4.1	−1.1 ± 3.3	0.47
**Secondary Outcomes**			
Total daily energy intake at final evaluation (kj)	7980 ± 3013 (*N* = 24)	8322 ± 1726 (*N* = 11)	0.73
Change in total daily energy between final and baseline evaluation (kj)	−1244 ± 3022 (*N* = 24)	83 ± 2975 (*N* = 11)	0.23
Daily fat intake at final evaluation (% total energy)	37 ± 9 (*N* = 24)	31 ± 4 (*N* = 11)	0.03
Daily saturated fat intake at final evaluation (% total energy)	11 ± 4 (*N* = 24)	9 ± 3 (*N* = 11)	0.15
Daily carbohydrate intake at final evaluation (% total energy)	39 ± 10 (*N* = 24)	48 ± 6 (*N* = 11)	0.008
Fibre intake at final evaluation (g/day)	24 ± 12 (*N* = 24)	30 ± 8 (*N* = 11)	0.14
Total weekly activity time at final evaluation (min)	209 ± 265 (*N* = 29)	142 ± 130 (*N* = 13)	0.40
7-day pedometer step count	46258 ± 29189 (*N* = 26)	39658 ± 16369 (*N* = 11)	0.40

**Table 3 nutrients-11-00590-t003:** Intervention group feedback regarding the text messages.

Feedback Question	Agree or Strongly Agree	Neutral	Disagree or Strongly Disagree
I found the texts useful	23	2	1
The majority of SMS were easy to understand	26	0	0
The text sms motivated me to change my lifestyle	20	4	2
As a result of the sms, my diet became more healthy	14	10	2
As a result of the messages, I increased my exercise (physical activity) levels	15	8	3
The sms helped remind me to have glucose tolerance test	18	4	3

sms = text message.

**Table 4 nutrients-11-00590-t004:** Predictors of activity monitor use amongst intervention subjects.

Variable	Used Activity Monitor at Least 50% of Days (*N* = 15)	Used Activity Monitor Less Than 50% of Days (*N* = 25)	*p* Value
Age > 34 years	8 (53%)	12 (48%)	1.0
Overweight or obese (BMI ≥ 25 kg/m^2^)	10 (67%)	19 (76%)	0.72
First degree family history of diabetes	9 (60%)	12 (48%)	0.53
Primiparous	2 (13%)	10 (40%)	0.15
South Asian ethnicity	8 (53%)	14 (56%)	1.0
Insulin use during pregnancy	9 (60%)	20 (80%)	0.27
Baseline energy intake exceeded recommendations	2 (17%)	10 (40%)	0.15
Baseline moderate intensity physical activity >150 min/week	6 (40%)	14 (56%)	0.51

Age 34 years = 50th percentile for whole cohort.

**Table 5 nutrients-11-00590-t005:** Intervention group feedback regarding the activity monitor.

Feedback Question	Agree or Strongly Agree	Neutral	Disagree or Strongly Disagree
I found the Fitbit useful	22 (85%)	2 (8%)	2 (8%)
I wore it most of the time	24 (96%)	1 (4%)	0 (0%)
I looked at my results most days	22 (85%)	3 (12%)	1 (4%)
Using the Fitbit motivated me to change my lifestyle	16 (62%)	8 (31%)	2 (8%)
As a result of the Fitbit, I increased my exercise (physical activity) levels	16 (62%)	8 (31%)	2 (8%)
I found the Fitbit easy to use and synchronise	18 (70%)	6 (23%)	2 (8%)

**Table 6 nutrients-11-00590-t006:** Predictors of study completion.

Variable	Completed All Elements of 6-Month Evaluation (*N* = 27)	Failed to Complete all Elements of 6-Month Evaluation (*N* = 33)	*p* Value
Intervention arm	19 (70%)	21 (64%)	0.79
Age > 34	16 (59%)	14 (42%)	0.30
Overweight or obese (BMI ≥ 25)	23 (85%)	23 (70%)	0.22
First degree family history of diabetes	14 (52%)	18 (54%)	1.0
Primiparous	3 (11%)	15 (46%)	0.005
South Asian ethnicity	11 (41%)	23 (70%)	0.04
Insulin use during pregnancy	20 (74%)	26 (79%)	0.76
Baseline energy intake exceeded recommendations	11 (41%)	16 (59%)	0.79
Baseline moderate intensity physical activity >150 min/week	11 (41%)	22 (67%)	0.07

Age 34 years = 50th percentile for whole cohort.

**Table 7 nutrients-11-00590-t007:** Sample size calculations based on outcomes of SMART MUMS WITH SMART PHONES.

Parameter	Sample Size Required
Total activity time	378
Step count	488
Change in body weight	976
Total energy intake	1892

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
