# Peer review of "A Pilot Randomised Controlled Trial of a Text Messaging Intervention with Customisation Using Linked Data from Wireless Wearable Activity Monitors to Improve Risk Factors Following Gestational Diabetes"

_nutrients, 2019, doi:10.3390/nu11030590_

Reviewer 1 Report

This is an interesting pilot study based around the use of modern phone technology to advise women at high risk of developing type 2 diabetes following a pregnancy affected by gestational diabetes, about strategies that may reduce their risk.

There are two main concerns that I have about the study. The main one is that I do not believe that this pilot study suggests that a larger trial would be very successful due to the fact that such a low proportion (<50%) of both intervention and control groups completed all elements of the study. Indeed 77% of the study participants failed to complete all elements of the study. This would be enough to put me off trying to run a more substantive trial without major modifications of the protocol.

My other main concern is that I disagree with the authors' interpretation of what measures displayed "trends" (Table 2). These were evident for (the secondary outcomes) daily fat intakes being higher in the intervention group, and dietary carbohydrate intake being lower in this group (without a change in overall energy intake being evident). No other measures showed trends, but this may relate to the low statistical power of the study arising from such the small number of participants that completed all elements of the study. I also disagree with the interpretation of data presented in Table 4 where there are no apparent tendencies.

There are also some other more minor points that I feel need addressing:

1. Why was this sample size chosen? It is fine for a pilot but I would like to see justification of it.

2. The study participants may be biased relative to the whole population through the inclusion criteria: mainly having to own a smart mobile phone and have internet access. What proportion of the local population would have been excluded by these inclusion criteria?

3. Do all women in the locality with diabetes in pregnancy attend this tertiary hospital antenatal clinic?

4. What proportion of women that were approached to enter the study were actually recruited to it?

5. In Table 1 please provide p-values for the comparison between the two groups just to confirm that there was no difference. Also please state what the data represent (numbers, means, standard deviations etc.).

6. In Table 2, if the measures of spread are standard deviations, some of the data appears to be normally distributed (where the mean is less than half the standard deviation in variables that can not be negative). How were these data dealt with statistically?

Author Response

This is an interesting pilot study based around the use of modern phone technology to advise women at high risk of developing type 2 diabetes following a pregnancy affected by gestational diabetes, about strategies that may reduce their risk.

There are two main concerns that I have about the study. The main one is that I do not believe that this pilot study suggests that a larger trial would be very successful due to the fact that such a low proportion (<50%) of both intervention and control groups completed all elements of the study. Indeed 77% of the study participants failed to complete all elements of the study. This would be enough to put me off trying to run a more substantive trial without major modifications of the protocol.

An important overall message is the importance of addressing feasibility in these studies and we agree that a new study would require modifications to address these feasibility issues.

We agree that the low completion rate is a concern, and in the paper we have discussed how a number of other studies in this field have also suffered from this problem. We have stated in the concluding paragraph that: “The repeatedly documented challenges of conducting behavioural intervention trials for this population and collection of high quality data, along with the large sample size and long duration of follow-up required to demonstrate a reduction in the rate of development of diabetes, raises the question as to whether further trials should be undertaken.”

We have now also changed this sentence “Our pilot study has demonstrated feasibility of a healthy lifestyle intervention based around text messaging and activity monitors for women with recent GDM, and lessons have been learnt which may facilitate the development of a future digital health support program” to “Our pilot study has demonstrated that a healthy lifestyle intervention based around text messaging and activity monitors is suitable for some women with recent GDM, but major protocol modifications are required to improve completion rates for a larger trial to be feasible. A number of lessons have been learnt which may facilitate the development of a future digital health support program.”

My other main concern is that I disagree with the authors' interpretation of what measures displayed "trends" (Table 2). These were evident for (the secondary outcomes) daily fat intakes being higher in the intervention group, and dietary carbohydrate intake being lower in this group (without a change in overall energy intake being evident). No other measures showed trends, but this may relate to the low statistical power of the study arising from such the small number of participants that completed all elements of the study. I also disagree with the interpretation of data presented in Table 4 where there are no apparent tendencies.

We have removed the interpretation from the results. This sentence: “However there were trends in favour of intervention for energy intake, healthier diet, weekly activity time, 7-day step count and self-reported body weight (table 2).” has been modified to only state the significant results: “However intervention subjects had a higher fat and lower carbohydrate intake, as percentages of total energy intake (table 2).” We agree that the problem is that we have inadequate statistical power both from a modest sample size and low completion rates.

We accept the reviewer’s concerns regarding interpretation of table 4 and have deleted this statement: “but irregular users tended to have a diet which exceeded recommended energy requirements at baseline and more often be primiparous (both p=0.15)”, leaving only “There were no factors which predicted activity monitor use.”

 There are also some other more minor points that I feel need addressing:

1. Why was this sample size chosen? It is fine for a pilot but I would like to see justification of it.

The sample size was actually dictated by the limited funding available. Essentially we were able to purchase 40 Fitbits and fund our dietitian and research coordinator within the budget.  We had no data upon which to base an effect size calculation prior to this study. We do not believe that it is normal practice to give specific funding details for a pilot study, but if the reviewer feels that it is important, we would be pleased to add in the text: “The sample size was determined by the size of the funding grant of A$50,000”.

 2. The study participants may be biased relative to the whole population through the inclusion criteria: mainly having to own a smart mobile phone and have internet access. What proportion of the local population would have been excluded by these inclusion criteria?

Uptake of mobile phone technology in this age group is virtually universal in Australia. The vast majority of those with mobile phones would have a smart phone, though we have not specifically surveyed this. We now indicate under “Participants and Randomization”, that “In 2018, 83% of Australian adults owned a smart phone [29], and the vast majority of women attending our clinics own smart phones.”

3. Do all women in the locality with diabetes in pregnancy attend this tertiary hospital antenatal clinic?

Women in our health district who chose to be treated in a public hospital for free attend our clinics as there is no other free option. There are some women who are managed privately and therefore are not seen in the clinics. We now indicate under “Participants and Randomization” that “All women in our health district who choose to be treated in a public hospital for free attend our clinics.”

 4. What proportion of women that were approached to enter the study were actually recruited to it?

We did not collect this data. It would have been difficult to accurately count the numbers as in some instances, various staff members approached women during the course of their normal consultations, and then referred those who were interested to our trial coordinator. However we now indicate how many women are seen by our service on an annual basis: “Our tertiary level hospital cares for about 900 women with GDM per annum.”

 5. In Table 1 please provide p-values for the comparison between the two groups just to confirm that there was no difference. Also please state what the data represent (numbers, means, standard deviations etc.).

In table 1 and table 2 we now indicate that the data are presented as mean±SD or proportion (%). There were no significant differences between intervention and control subjects. We have now added an extra column to show individual p values in table 1.

 6. In Table 2, if the measures of spread are standard deviations, some of the data appears to be normally distributed (where the mean is less than half the standard deviation in variables that can not be negative). How were these data dealt with statistically?

All data were normally distributed except for the measures of physical activity. However whether the data were analysed by t-tests or the Mann Whitney U test, the differences were not significant. As this was the case, we decided to be consistent in the table by showing mean±SD for weekly activity time and pedometer step count and the results of the t-test, rather than as median with IQR and explain the need for different statistical tests just for these 2 outcomes. For the interest of the reviewer, the p value for total weekly activity time at final evaluation was 0.83 when analysed by Mann Whitney U, and for 7-day pedometer step count the p value was 0.73.

Reviewer 2 Report

In this pilot RCT by Cheung NW et al they adopted two forms of m-health technology, text messaging, and wireless web-linked wearable activity monitors, and linked them to test a novel intervention, which supports lifestyle change amongst women with recent GDM to reduce diabetes risk amongst these women. Sixty subjects were included and randomised 2:1 intervention vs control. They demonstrated a trend to improvement in physical activity, dietary energy intake and body weight.

Comments:

This is a very interesting study as the incidence of GDM is increasing all over the world and the future risk of type 2 diabetes in this group is now a major population health issue. The design is straightforward with a low resource intervention using the new technology. Unfortunately, the results are disappointing. There were no difference between the control and intervention groups regarding dietary and physical activity targets achieved. Furthermore, there was a very low completion rate, which is a concern for future studies in this area.

Once again, this study shows the problems that are involved in lifestyle interventions.

Specific points:

Regarding recruitment, it is striking that 73% and 85% respectively use insulin during pregnancy. That seems very high, is that normal for Australian GDM patients?

There was no difference between the groups in the rate of glucose tolerance testing (GTT), but what was the result of the GTT performed 12 weeks post-partum? I assume that the “Pregnancy GTT” in Table 1 is the GTT performed during pregnancy.  

As the study concerns diabetes, why wasn’t “Daily carbohydrate intake” one of the primary outcomes? According to Table 2, “Daily carbohydrate intake at final evaluation” is one of only two significant result in this study.

Line 364: It is stated that; “there was a shift from the higher carbohydrate requirements of pregnancy, to a more modest carbohydrate intake post-partum” I cannot find the daily carbohydrate intake at baseline or pregnancy.

In table 2; Diet fat intake<30% of total energy at final evaluation was 17% (Intervention group 4/24) and 36 % (Control 4/11) while the Daily fat intake at final evaluation (% total energy) was 37±9 (N=24) 31±4 (N=11) which was significantly lower. How come? Was it because the seven (11-4=7) controls ate very much fat? The same goes for saturated fat.

Why a “?” after “GTT done?” in figure 1?

Author Response

In this pilot RCT by Cheung NW et al they adopted two forms of m-health technology, text messaging, and wireless web-linked wearable activity monitors, and linked them to test a novel intervention, which supports lifestyle change amongst women with recent GDM to reduce diabetes risk amongst these women. Sixty subjects were included and randomised 2:1 intervention vs control. They demonstrated a trend to improvement in physical activity, dietary energy intake and body weight.

Comments:

This is a very interesting study as the incidence of GDM is increasing all over the world and the future risk of type 2 diabetes in this group is now a major population health issue. The design is straightforward with a low resource intervention using the new technology. Unfortunately, the results are disappointing. There were no difference between the control and intervention groups regarding dietary and physical activity targets achieved. Furthermore, there was a very low completion rate, which is a concern for future studies in this area.

Once again, this study shows the problems that are involved in lifestyle interventions.

Specific points:

Regarding recruitment, it is striking that 73% and 85% respectively use insulin during pregnancy. That seems very high, is that normal for Australian GDM patients?

The subjects were mainly recruited from our multidisciplinary Diabetes in Pregnancy antenatal clinic, where women needing insulin are seen. Women who do not require insulin are seen by diabetes educators, but do not attend this clinic. Under “Participants and Randomization”, we have added “Women attending this clinic were largely those who required insulin, as women not needing insulin were seen in regular antenatal clinics.”

 There was no difference between the groups in the rate of glucose tolerance testing (GTT), but what was the result of the GTT performed 12 weeks post-partum? I assume that the “Pregnancy GTT” in Table 1 is the GTT performed during pregnancy.  

We now include the post-partum GTT results in the text, but as this was not a prespecified outcome of interest, we have not included it in table 2. On the post-partum GTT, there was one woman from each group who met the criteria for diabetes. We have added the following text in the first paragraph of the results: “One woman from each arm of the study had developed diabetes on the GTT. The glucose values on the GTT were not different between intervention and control subjects (fasting glucose 4.9±0.7 vs 5.0±4.2 mmol/L, p=0.9; and 2 hour glucose 7.2±2.2 vs 6.4±2.0 mmol/L, p=0.2)”. We have also added in the methods: “For the women who had developed diabetes, clinical management was left with the woman’s primary practitioner, and the research team did not become involved.”

 As the study concerns diabetes, why wasn’t “Daily carbohydrate intake” one of the primary outcomes? According to Table 2, “Daily carbohydrate intake at final evaluation” is one of only two significant result in this study.

Carbohydrate intake outside of pregnancy is a controversial area. The Australian Guidelines for Healthy Eating recommend that women in this age group consume a minimum of 6 serves of grains a day, with no specific maximum for carbohydrate. It is also recommended that the carbohydrate should predominantly be of whole grain or high fibre cereal. With the Australian guidelines in mind, we did not set a target amount of carbohydrate as a primary outcome. However through our counselling sessions, we promoted moderation of carbohydrate intake and an emphasis on low GI carbohydrate.

 Line 364: It is stated that; “there was a shift from the higher carbohydrate requirements of pregnancy, to a more modest carbohydrate intake post-partum” I cannot find the daily carbohydrate intake at baseline or pregnancy.

We have now included carbohydrate intake at baseline in table 1. The percentage of total daily energy consumed as carbohydrate in the intervention group was 43±10 and for controls 47±8. There was no difference between them (p=0.34).

 In table 2; Diet fat intake<30% of total energy at final evaluation was 17% (Intervention group 4/24) and 36 % (Control 4/11) while the Daily fat intake at final evaluation (% total energy) was 37±9 (N=24) 31±4 (N=11) which was significantly lower. How come? Was it because the seven (11-4=7) controls ate very much fat? The same goes for saturated fat.

17% (4/24) intervention subjects met the standard of dietary fat<30% and="" therefore="" conversely="" consumed="">30%. Amongst the control subjects 36% (4/11) met the standard of dietary fat<30% and="" therefore="" consumed="">30%. Thus both these results indicate that intervention subjects consumed more fat, as a percent of total energy intake, consistent with the 37±9 for intervention and 31±4 for controls. The explanation is the same for saturated fat.

 Why a “?” after “GTT done?” in figure 1?

We now clarify that we ascertained if the GTT had been performed by changing the text in the figure to “Was GTT done?”

Round  2

Reviewer 1 Report

I like the revisions that the authors have made to their manuscript, and feel that they have improved the quality of it. In particular I like how the authors have dealt with my main concern, relating to the large drop out of participants, in the Discussion section. However the abstract was not modified and I feel that its final sentence needs to be changed to reflect the revised Discussion section.

Author Response

We agree that it is appropriate to modify the end of the abstract. The last sentences now read

"Overall, results suggest that a text message and activity monitor intervention is feasible for a larger study or even as a potentially scalable population health intervention. However low completion rates necessitate carefully considered modification of the protocol."